# A Robust Power Allocation Algorithm for Cognitive Radio Networks Based on Hybrid PSO

**DOI:** 10.3390/s22186796

**Published:** 2022-09-08

**Authors:** Lu Zhao, Mingyue Zhou

**Affiliations:** College of Computer Science and Technology, Changchun University of Technology, Changchun 130012, China

**Keywords:** cognitive radio, robust optimization, resource allocation, particle swarm algorithm

## Abstract

The use of a cognitive radio power allocation algorithm is an effective method to improve spectral utilization. However, there are three problems with traditional cognitive radio power allocation algorithms: (1) based on the ideal channel model analysis, channel fluctuation is not considered; (2) they do not consider fairness among cognitive users; and (3) some algorithms are complex and locating the optimal power allocation scheme is not an easy task. For the above problems, this study establishes a robust model which adds the cognitive user transmission rate variance constraint to solve the maximum channel capacity time power allocation scheme by considering the worst-case channel transmission model, and finally solves this complex non-convex optimization problem by using the hybrid particle swarm algorithm. Simulation results show that the algorithm has good robustness, improves the fairness among the cognitive users, makes full use of the channel resources under the constraints, and has a simple algorithm, fast convergence, and good optimization results.

## 1. Introduction

Presently, various wireless communication services are experiencing exponential growth. For example, the 5G mobile communication networks [1] will connect 100 billion devices worldwide. SpaceX’s “chain plan” [2] will launch 42,000 LEO communication satellites into space. To address the issue of spectrum resource shortage of these huge numbers of communication services, it is necessary to introduce unauthorized spectrum access [3], and also to ensure a certain degree of fairness. In this context, cognitive radio (CR) [4,5,6] technology has become the focus of wireless communication research. CR technology can dynamically access spectrum resources and improve the utilization of spectrum resources.

### 1.1. Previous Studies

Many papers consider the power allocation problem based on the perfect channel state information. In practice, some system parameters are unstable. To solve this problem, authors in [7] considered the distributed power allocation algorithm in CR networks, and ultimately addressed the system’s robust power control issue using the dual decomposition theory. In [8], authors studied a CR network in which multiple primary users work at the same time, and solved the problem of ideal channel state by combining semidefinite relaxation with particle swarm optimization. In order to improve the fairness among cognitive users. Authors in [9] examined inter network collaboration for security aware joint power and subchannel allocation in cognitive heterogeneous networks, improving throughput and proportionate justice among cognitive users. Authors [10] have proposed an effective resource allocation plan and dynamic MAC frame configuration for a multi-channel ad-hoc CR network. The particle swarm optimization (PSO) [11] technique converts the dynamic resource allocation model into a restricted optimization problem of a multi-objective function, which optimized the objective function and offered proportionate justice among the cognitive users. Authors [12] studied multi-user resource allocation in the orthogonal frequency division multiplexing (OFDM) CR network. The power allocation is carried out through PSO algorithm, which not only improves the fairness among cognitive users, but also reduces the complexity of the algorithm. Authors [13], proposed a hybrid algorithm of PSO and gravitational search algorithm (GSA), and realized the importance of balance between the exploration and development ability of PSO-GSA algorithm. This algorithm can detect spectrum holes with optimal transmission power, sensing bandwidth and power spectral density, thus improving the energy efficiency of spectrum sensing. References [14,15,16,17,18,19,20,21,22,23] have conducted various improvement studies on the PSO algorithm to enhance the PSO algorithm’s capacity for optimization and convergence.

### 1.2. Main Content and Arrangement

Previous studies have found that some of the resource allocation algorithms in the underlying CR network are not robust, some lack fairness among cognitive users, or the algorithms are too complex. Therefore, this paper investigates a robust model [24,25] which underlies CR networks with the objective of maximizing channel capacity [26] and constraints of cognitive user power, primary user interference threshold, and transmission rate variance of cognitive users. The research contents on this model are as follows: (1) establish a robust model to improve the transmission rate variance constraint of cognitive users in order to improve the fairness of cognitive users; (2) a collection of ellipsoids [27] is used to describe the system’s uncertainty, and the semi-infinite planning problem (SIP) [7] is transformed into a finite number of constrained problems that can be solved by considering the worst-case system planning; and (3) study of a hybrid PSO algorithm. The adaptive weighting method [28] can make the inertia weights of particles adjust according to the fitness, while the natural selection algorithm [29] can filter out the fitness particles for the next iteration. Together, the two techniques strengthen the particle swarm algorithm’s capacity for search and convergence.

The remaining structure of this essay is as follows: in Section 2, the system model is presented. The robust model is introduced in Section 2.1. The hybrid PSO power allocation algorithm is introduced in Section 3. The experimental simulation analysis is carried out in Section 4, and Section 5 contains the conclusion.

## 2. System Model

In the ad-hoc [10] distributed CR network, in which there is no central control node, we assume that there are S cognitive user links are randomly assigned in the cluster area, as shown in Figure 1. Where SUs represents the cognitive user, PU represents the primary user and PU-T represents the primary user transmitter. PU-R represents the primary user receiver. SUs-T represents the cognitive user s transmitter, while SUs-R is the cognitive user s receiver. The dotted line denotes the interfering connection. $h_{s}$ denotes the channel gain of the link from SUs-T to PU-R, $G_{ss}$ is the channel gain from SUs-T to SUs-R, and $g_{0}$ represents the channel gain from PU-T to SU-R.

This model is an underlay model (i.e., the cognitive user can use the spectrum resources at the same time as the primary user), but the interference from the cognitive user to the primary use must be less than the interference threshold of the primary user. That is, the sum of the interference power of the SU-T cannot be greater than the interference threshold of the PU-R, as shown in Equation (1) below.
(1)$$\sum_{i=1}^{m}h_{i}p_{i}\leq I$$

Here $p_{i}$ means the transmit power of SUi-T and $h_{i}$ is the channel gain among SU-T and PU-R of link i. As well as I indicates the interference threshold that the primary user can tolerated.

Secondly, each SU-T must have a transmit power that is either below or comparable to its maximum transmit power, which can be expressed by the following expression (2):(2)$$p_{i}\leq p_{\max}$$
where $p_{\max}$ is the maximum transmission power of SUs.

In the pursuit of certain transmission fairness for cognitive users, traditional methods achieve fairness by setting the proportion of SUs’ transmission rate. This method is complicated, and the transmission power limit is too strict, which leads to the channel capacity loss of the single machine. Therefore, in this paper, the channel capacity variance of each SUs is the constraint condition, so as not to limit the transmission power of the unit too much, as to give full use of CR resources. This not only improves the fairness of the whole system, but also makes the algorithm simpler and more flexible than the proportional transmission rate algorithm. The SUs transmission rate variance constraint are as follows:(3)$$C_{i}=\log_{2}(1+\frac{G_{ii}p_{i}}{\sum_{j\neq i}^{m}G_{ij}p_{j}+g_{0}p_{0}+\eta })$$
(4)$$\bar{C}=\frac{1}{m}\sum_{i=1}^{m}C_{i}$$
(5)$$\frac{\sum_{i=1}^{m}{\left(C_{i}-\bar{C}\right)}^{2}}{m}\leq Dx$$

In the Formula (3), $C_{i}$ is the transmission speed of cognitive user i. $G_{ii}$ is the channel gain from SUi-T to SUi-R. $G_{ij}$ is the channel gain from SUj-T to SUi-R. $g_{0}$ represents the channel gain from PU-T to SUs-R. $p_{0}$ is the PU transmit power. η is the background noise of this system model. In Formula (4), $\bar{C}$ represents the average transmission rate of SUs. Lastly, in Formula (5), $D_{x}$ is the transmission rate variance threshold.

The objective of this model is to increase channel capacity. Combining the constraints (1), (2), and (5), the ideal channel model is as follows:(6)$$\begin{array}{l} \max\sum_{i=1}^{m}\log_{2}(1+\frac{G_{ii}p_{i}}{\sum_{j\neq i}^{m}G_{ij}p_{j}+g_{0}p_{0}+\eta })\\ \begin{array}{lll} s.t. & C1: & \sum_{i=1}^{m}h_{i}p_{i}\;\leq \;I\\ & C2: & p_{i}\leq p_{\max}\\ & C3: & \frac{\sum_{i=1}^{m}{\left(C_{i}-\bar{C}\right)}^{2}}{m}\leq Dx \end{array} \end{array}$$

### 2.1. Robust Model

Most power allocation algorithms in CR networks are based on perfect channel estimation [21], but in practical applications, background noise and channel gain of CR systems fluctuate slightly due to environmental impacts. In order to ensure that the communication of the whole system is not interrupted, a robust algorithm is adopted. The best advantage of this algorithm is that solutions in unstable systems can remain relatively stable. Therefore, this paper considers the worst channel model, sacrificing a small part of the system efficiency to achieve the stability of the system.

The channel interference gain is expressed as follows:(7)$$\sigma_{ij}=\{\begin{array}{c} \frac{G_{ij}}{G_{ii}},i\neq j,\\ 0,i=j, \end{array}$$
(8)$$\sigma_{ij}=\bar{\sigma_{ij}}+\Delta \sigma_{ij}$$

Divide channel interference gain into two parts: $\bar{\sigma_{ij}}$ indicates the nominal interference gain, and $\Delta \sigma_{ij}$ indicates the perturbation of interference gain.

Similarly, $h_{i}$ can also be divided into two parts: $\bar{h_{i}}$ indicates the nominal interference gain of SUi-T to PU-R, and $\Delta h_{i}$ indicates the perturbation of interference gain of SUi-T to PU-R. As shown in the following Formula (9):(9)$$h_{i}=\bar{h_{i}}+\Delta h_{i}$$

The description of uncertainty sets is also a difficult point of robust optimization. The difficulty of solution will be greatly increased for too fine uncertainty sets, and the optimal solution obtained for too broad uncertainty sets will be too conservative. In this paper, the classical ellipsoidal uncertainty sets are used to solve the problem. When dealing with linear programming, quadratic optimization, and cone quadratic optimization, the method can transform the problem into a tractable problem. The specific formula is as follows (10):(10)$$U=\left\{\xi :{\left\|\xi \right\|}_{2}\leq \Omega \right\}=\left\{\xi :\sum \xi_{i}^{2}\leq \Omega^{2}\right\}$$
where U is the expected or predicted value vector of the uncertain parameter, ξ is the uncertain parameter vector, and Ω is the uncertainty, which is used to describe the disturbance range of the uncertain parameter, using an elliptic set to describe the uncertainty of channel interference gain, as shown in Equation (11):(11)$$\delta_{i}=\left\{\bar{\sigma_{i}}+\Delta \sigma_{i}:\sum_{i\neq j}{\left|\Delta \sigma_{ij}\right|}^{2}\leq {\left(\varepsilon_{i}\right)}^{2}\right\}$$

In the above Equation (10), $\bar{\sigma_{i}}$ denotes the nominal part of interference gain to SUi-R, $\Delta \sigma_{i}$ represents the perturbation of interference gain to SUi-R, ${\left(\varepsilon_{i}\right)}^{2}$ is the maximum acceptable deviation of channel gain, i.e., the upper limit of the sum of squares of the perturbation $\Delta \sigma_{ij}$ of the interference gain.

Similarly, the cognitive user’s primary channel interference gain can also be represented by an elliptic set as follows:(12)$$H=\left\{\bar{h}+\Delta h:\sum_{i}{\left|\Delta h_{i}\right|}^{2}\leq \varepsilon_{0}^{2}\right\}$$
where H represents the uncertainty set of the interference channel gain of the SU-T to PU-R; $\bar{h}$ is the nominal part of the interference channel gain, Δh represents the perturbation of interference channel gain; and $\varepsilon_{0}^{2}$ is the square of the maximum acceptable deviation of h

Thus, the above (6) model is transformed into the following robust model:(13)$$\begin{array}{l} \max\sum_{i=1}^{m}\log_{2}\left(1+\frac{p_{i}}{\sum_{j\neq i}\left(\bar{\sigma_{ij}}+\Delta \sigma_{ij}\right)p_{j}+\left(g_{0}p_{0}+\eta \right)/G_{ii}}\right)\\ \begin{array}{lll} s.t. & C1: & \sum_{i=1}^{m}\left(\bar{h_{i}}+\Delta h_{i}\right)p_{i}\leq I\\ & C2: & p_{i}\leq p_{\max}\\ & C3: & \frac{\sum_{i=1}^{m}{\left(C_{i}-\bar{C}\right)}^{2}}{m}\leq Dx \end{array} \end{array}$$

In the above formula, $C_{i}$ can be rewritten as follows:(14)$$\begin{array}{l} C_{i}=\log_{2}(1+\frac{p_{i}}{\sum_{j\neq i}^{m}\left(\bar{\sigma_{ij}}+\Delta \sigma_{ij}\right)p_{j}+\left(g_{0}p_{0}+\eta \right)/G_{ii}}) \end{array}$$

The robust optimization problem is a SIP problem. In this model, due to the fluctuating channel gain between cognitive users, and the fluctuating channel gain between cognitive and primary users, there are infinite possibilities in a small range, resulting in infinite possibilities for the constraints associated with them. Therefore, the use of ellipsoidal sets to describe the uncertainty of the system does not solve this problem, but merely describes the uncertainty of the system through mathematical expressions to facilitate further processing. There is no direct solution to this SIP problem. We must transform a problem with infinite constraints into a problem with finite constraints. By considering the above in the worst-case scenario, the following equation is obtained through Cauchy Schwarz inequality [7].
(15)$$\underset{\sigma_{i}\in \delta_{i}}{\max}\left\{\sum_{j\neq i}\left(\sigma_{ij}-\bar{\sigma_{ij}}\right)p_{j}\right\}=\varepsilon_{i}\sqrt{\sum_{j\neq i}{\left(p_{j}\right)}^{2}}$$
(16)$$\underset{h\in H}{\max}\left\{\sum_{i}\left(h_{i}-\bar{h_{i}}\right)p_{i}\right\}=\varepsilon_{0}\sqrt{\sum_{i}{\left(p_{i}\right)}^{2}}$$

We can get the following robust model:(17)$$\begin{array}{l} \max\sum_{i=1}^{m}\log_{2}\left(1+\frac{p_{i}}{\sum_{j\neq i}\bar{\sigma_{ij}}p_{j}+\varepsilon_{i}\sqrt{\sum_{j\neq i}{\left(p_{j}\right)}^{2}}+\left(g_{0}p_{0}+\eta \right)/G_{ii}}\right)\\ \begin{array}{lll} s.t. & C1: & \sum_{i=1}^{m}\left(\bar{h_{i}}+\varepsilon_{0}\sqrt{\sum_{i}{\left(p_{i}\right)}^{2}}\right)p_{i}\leq I\\ & C2: & p_{i}\leq p_{\max}\\ & C3: & \frac{\sum_{i=1}^{m}{\left(C_{i}-\bar{C}\right)}^{2}}{m}\leq Dx \end{array} \end{array}$$

## 3. Hybrid PSO Algorithm for Power Allocation

This paper uses PSO to solve the optimization model. The algorithm has the advantages of fast convergence and strong search ability. In addition, the algorithm can continue to develop optimization algorithms such as hybrid algorithm and simulated annealing algorithm to improve the global search ability, accuracy, and convergence speed of PSO algorithm.

Therefore, in order to improve the algorithm’s search ability, adaptive weighting and natural selection algorithm are added to the conventional PSO algorithm.

### 3.1. PSO with Constraints

In the particle swarm algorithm, there are N particles flying in D dimension space, and their position coordinates in flight represent every possible solution of the optimization model. Fitness functions are used to determine if a location is good. A position with a good fitness value indicates that the current position is good, and a position with a poor fitness value indicates that the current position is bad. The speed and position of these particles are adjusted by global extremes (historical optimal fitness values for the entire population) and individual extremes (historical optimal fitness values found by the particles themselves). These particles cooperate by sharing information global extremum. Eventually, all the particles will fly to the same location, which is ultimately the most optimal solution. The velocity and position of the particles are updated by two formulas:(18)$$v_{id}^{t+1}=w\times v_{id}^{t}+c_{1}r_{1}\left(p_{id}^{t}-x_{id}^{t}\right)+c_{2}r_{2}\left(p_{gd}^{t}-x_{id}^{t}\right)$$
(19)$$x_{id}^{t+1}=x_{id}^{t}+v_{id}^{t+1}$$
w is inertia weight. $x_{id}^{t}$ and $v_{id}^{t}$ respectively represent the location and speed of particle i in the D-dimensional space at time *t*. $c_{1}$ is the self-learning factor and $c_{2}$ is the group learning factor. The random numbers $r_{1}$ and $r_{2}$ range from 0 to 1. $p_{id}^{t}$ represents the individual extreme value and $p_{gd}^{t}$ represents the global extreme value.

In the search space, some locations belong to the infeasible domain due to the existence of restrictions. A penalty function is added to guide the particles to the feasible domain and search for the optimal solution in the feasible domain. The fundamental concept is to make particles’ fitness invariant when they meet the constraint conditions, and if the restriction is not satisfied, punish it so that its fitness is extremely poor, and then adjust its position and speed to jump out of the infeasible area.
(20)$$\begin{array}{l} P\left(x\right)=fitness\left(x\right)+\sum_{i}p_{i}\left(x\right)\\ \{\begin{array}{c} p_{1}\left(x\right)=\{\begin{array}{cc} 0 & p_{1p}\left(x\right)\leq 0\\ -{\left(10\right)}^{10} & p_{1p}\left(x\right)>0 \end{array}\\ p_{2}\left(x\right)=\{\begin{array}{cc} 0 & p_{2p}\left(x\right)\leq 0\\ -{\left(10\right)}^{10} & p_{2p}\left(x\right)>0 \end{array}\\ p_{3}\left(x\right)=\{\begin{array}{cc} 0 & p_{3p}\left(x\right)\leq 0\\ -{\left(10\right)}^{10} & p_{3p}\left(x\right)>0 \end{array} \end{array} \end{array}$$

In (19), P(x) is the penalty function [24], fitness(x) is the fitness function, $p_{i}\left(x\right)$ is the penalty item, and $p_{1p}\left(x\right)$, $p_{2p}\left(x\right)$,$\;p_{3p}\left(x\right)$ are three constraint judgment functions, as following Formulas (21)–(24).
(21)$$fitness\left(x\right)=\sum_{i=1}^{m}\log_{2}\left(1+\frac{p_{i}^{x}}{\sum_{j\neq i}\bar{\sigma_{ij}}p_{j}^{x}+\varepsilon_{i}\sqrt{\sum_{j\neq i}{\left(p_{j}^{x}\right)}^{2}}+\left(g_{0}p_{0}+\eta \right)/G_{ii}}\right),\;x=\left\{1,\;2,\;\ldots N\right\}$$
(22)$$p_{1p}\left(x\right)=\sum_{i=1}^{m}\left(\bar{h_{i}}+\varepsilon_{0}\sqrt{\sum_{i}{\left(p_{i}^{x}\right)}^{2}}\right)p_{i}^{x}-I,\;x=\left\{1,\;2,\;\ldots N\right\}$$
(23)$$p_{2p}\left(x\right)=p_{i}^{x}-p_{\max},\;x=\left\{1,\;2,\;\ldots N\right\}$$
(24)$$p_{3p}\left(x\right)=\frac{\sum_{i=1}^{m}{\left(C_{i}-\bar{C}\right)}^{2}}{m}-Dx,\;x=\left\{1,\;2,\;\ldots N\right\}$$

### 3.2. Adaptive Weighting Method

The bigger the inertia weight in the PSO algorithm, the larger the search step in partial flight, and the stronger the capability of global search. As opposed to that, the smaller the inertia weight is, the smaller the search step in particle flight is, and the stronger the local search ability is. Therefore, appropriate inertia weight is very important [17].

The basic concept of adaptive weight method is as follows: a particle’s fitness is better than the average fitness, then its region has a high probability of global optimization, so the inertia weight should be reduced, and the local search ability increased. Whenever a particle’s fitness is worse than the global fitness, the probability of global optimization is low in the region where the region is located. Therefore, in order to improve the capacity of global search, the inertia weight must be increased.

Compare the particle’s present position fitness value to the overall fitness value. If particle’s present position’s fitness value is better than the average fitness value, the inertia weight will be reduced, and the local searching ability will be improved. If it is lower than the average fitness value, the inertia weight will be increased to quickly jump out of the region and improve the global searching ability [30].

The following formula is used to update the inertia weight:(25)$$\omega =\{\begin{array}{ll} \omega_{\max}-\frac{\left(\omega_{\max}-\omega_{\min}\right)\times \left(f-f_{\min}\right)}{f_{avg}-f_{\min}}, & f\geq f_{avg}\\ \omega_{\max}, & f<f_{avg} \end{array}$$

In the Formula (25), $\omega_{\max}$ and $\omega_{\min}$ indicate the max and min values of the inertia weights. Separately, f represents the current fitness value, and $f_{avg}$ represents the average fitness value. The inertia weight is decreased and the particle’s flying speed is slowed down when the particle’s fitness value exceeds or is equal to the current average fitness value. The particle’s inertia weight [28] is set to the maximum inertia weight when the particle’s fitness value is lower than the current average fitness value.

### 3.3. Natural Selection Algorithm

The algorithm’s central concept is in each iteration process, all particles are sorted according to their fitness, and then the particles with good fitness are used to replace the particles with poor fitness, keeping the original historical optimal value of each particle unchanged. This algorithm can greatly improve the convergence speed of PSO algorithm.

### 3.4. Hybrid PSO Algorithm

Hybrid PSO algorithm is a combination of adaptive weighting and natural selection. The specific algorithm steps are shown in the Algorithm 1 below:
**Algorithm 1**: Hybrid PSO algorithm for power allocation1. Set population parameters:Initialize the number of particles, search space, maximum number of iterations, inertia weight, self-learning factor, group learning factor, set location speed limit.2. Generate initial population:(1) Generate initial population location and speed.(2) Initialize individual and group history best position.(3) Initialize optimal fitness of individuals and groups. 3. Particle swarm iteration:(1) Update weights according to Formula (24).(2) Update location and speed, and boundary handling.(3) Make restriction judgments and calculate new fitness.(4) Conduct natural selection processing: the particles are sorted according to the fitness, and the half good position is used to replace the other half bad position. (5) Check to see whether the number of iterations is at its limit, then judge whether the output condition is reached. If the output result is reached, otherwise return to step 3.4. Output iteration results.

## 4. Results

In this section, we will demonstrate the performance of the algorithm through simulation experiments. Specific experimental parameters are shown in the Table 1 below.

Various algorithms are as follows: PSO represents the PSO method proposed in reference [10], Robust-PSO represents the PSO method proposed in reference [10] with robust algorithm, RAD-PSO represents the adaptive PSO method proposed in reference [17] with robust optimization, and RAN-PSO represents the natural selection + adaptive + robust hybrid PSO method. The parameters of various particle swarm optimization algorithms are shown in Table 2 below.

### 4.1. Performance Comparison

As shown in Figure 2, the four algorithms PSO, Robust-PSO, RAD-PSO and RAN-PSO are compared in terms of channel capacity convergence for the same population size, the same learning factor, and the same number of iterations. It can be seen that the PSO algorithm has the largest channel capacity, but this algorithm is not robust enough to be implemented in practice. In addition, this algorithm has a slow convergence rate. The Robust-PSO method has the lowest channel capacity, and its convergence rate is the slowest. The RAD-PSO algorithm has a better channel capacity and convergence rate than the Robust-PSO algorithm only. The channel capacity and convergence speed of the RAN-PSO algorithm are better than those of the RAD-PSO and robust PSO algorithms. While it is robust compared to PSO algorithm, the above also shows that the RAN-PSO algorithm converges faster, and has better channel capacity. The results show that the algorithm is effective, the robust algorithm provides robustness for the model, the adaptive weighting method improves the convergence speed and channel capacity of the algorithm, and the natural selection algorithm improves channel capacity further. In other words, the adaptive weighting method improves the global search capability and the local search capability, while the natural selection algorithm further improves the local search capability.

The following Figure 3a–d show the transmission rates of cognitive users for the four algorithms PSO, Robust-PSO, RAD-PSO, and RAN-PSO. In these figures, CR1, CR2, and CR3 denote cognitive users 1, 2, and 3, respectively. The four algorithms are compared for the fairness achieved among cognitive users in the model, and the comparison shows that (a) the transmission rate of three cognitive users in (a) has the largest dispersion, while the transmission rate of three cognitive users in (c) has the smallest dispersion, which means that the PSO algorithm has the worst fairness, while the RAD-PSO algorithm has the best fairness. As long as the constraints are met, the fairness requirement of cognitive users is not required to be good, because too much fairness will lead to the waste of channel resources. The fairness of the RAN-PSO algorithm is only better than the PSO algorithm, but the PSO algorithm is not robust. Taken together, we can conclude that the RAN-PSO algorithm has better performance and more accurate power allocation, and that RAN-PSO remains the fastest convergence algorithm in cognitive user transmission rate convergence.

Figure 4 shows the convergence process of channel capacity with different robustness coefficients. The robustness factor is used to describe the fluctuation of uncertain parameters. This is due to the fact that the larger the robustness factor, the larger the fluctuation of uncertain parameters. For the robust algorithm, the larger the robustness factor, the worse the state of the system considered. The robustness factor in this algorithm will make the model more constrained, and thus gain a higher robustness. It can be seen that algorithm channel with a robustness factor of 0.1 has the least capacity, while the algorithm channel with a robustness factor of 0.06 has the largest capacity. With the increase of the robustness factor, the channel capacity system decreases. It can be concluded that too high of a robustness factor leads to too conservative of an algorithm and too low of a robustness factor leads to too poor robustness of the system to be used in practice. Therefore, it is very important to find a balance between the robustness of the algorithm and the performance of the system.

Figure 5 shows the convergence process of channel capacity of the RAN-PSO algorithm in different population sizes. It can be observed that the RAN-PSO algorithm with a population size of 600 converges the fastest and starts to roughly stabilize at the sixth iteration. The RAN-PSO algorithm with 150 population sizes converged the slowest and starts to stabilize at the tenth iteration. The results show that the population size has a positive correlation with the convergence speed, i.e., the larger the population, the faster the convergence speed of channel capacity. The difference between the fastest convergence speed and the slowest convergence speed is not very large, which indicates that the convergence speed of this algorithm is usually very fast. The RAN-PSO algorithm with a population size of 600 has the maximum channel capacity, while the RAN-PSO algorithm with a population size of 150 has the minimum channel capacity. However, this does not mean that the larger the population size is better, because the RAN-PSO algorithm with a population size of 300 is better than the algorithm with a population size of 450 in terms of both convergence speed and channel capacity. However, the convergence speed and channel capacity of the population size of 600 are optimal, but this population size is too large when considering that, an algorithm with a population size of 300 is used in all experiments in this paper.

### 4.2. Other Performance of RAN- PSO Method

As shown in Figure 6, the change of transmit power of the three cognitive users during the iterative process is represented in (a), and the change process of SINR of the three cognitive users is shown in (b). In these figures, we can see that the shapes of the iterative curves of the two plots are very similar. However, CR3 has the largest transmit power as well as the largest SINR, while CR1 has the smallest transmit power and the smallest SINR. This phenomenon is due to the fact that in the process of cognitive radio power allocation, we pursue to maximize the channel capacity, and the algorithm will unconsciously assign a larger transmit power to the cognitive users with small channel interference gain and a smaller transmit power to the users with large interference channel gain, such that the users with poor channel conditions will cause less interference for the users with good channel conditions, and the users with good channel conditions can communicate better and improve the channel’s the utilization of resources. In other words, we focus our advantageous resources on users with good channel conditions to increase the channel capacity of the system. Of course, we do this under the premise of satisfying the constraints of cognitive user fairness, and we find that the SNR and transmission power of the three users in the figure are very close together and there are no extremes, so the fairness of the algorithm is guaranteed.

In Figure 7, we investigate the interference changes from cognitive users to primary users during the iteration of the RAN-PSO algorithm. The red line in the diagram indicates the interference threshold, and the black line with “*” indicates the interference of the cognitive user to the primary user. It can be seen that with iterative updates of the RAN-PSO algorithm, interference by cognitive users of primary users increases and gradually overlaps with the interference threshold. I This occurs only infinitely close to the interference threshold, as interference in constraints must be less than the interference threshold or be punished with a punitive function, resulting in extreme unsuitability, which is then eliminated. The interference of cognitive users to primary users is increasing, which means that the channel capacity of cognitive users is still increasing, and the algorithm improves channel resource utilization under this constraint. From the point of view of the algorithm, the RAN-PSO algorithm has strong local search capability and can obtain more accurate optimal solutions.

## 5. Conclusions

A robust power allocation algorithm for opportunity hybrid particle swarm algorithms is proposed with the objectives of solving fairness among cognitive users, resolving channel uncertainty, and reducing the difficulty of the algorithm in ad-hoc underlay CR networks. The uncertainty of the system is obtained by means of ellipsoidal sets, and the problem with infinite constraint is equivalent to the problem with finite constraint in the Cauchy Schwartz inequality. In the hybrid PSO algorithm, the constraint is dealt with by penalty functions, which reduce the difficulty of the algorithm.

Finally, the method’s performance is verified through simulation experiments in terms of channel capacity and SINR of cognitive users, as well as interference with primary users.

## Figures and Tables

**Figure 1 sensors-22-06796-f001:**
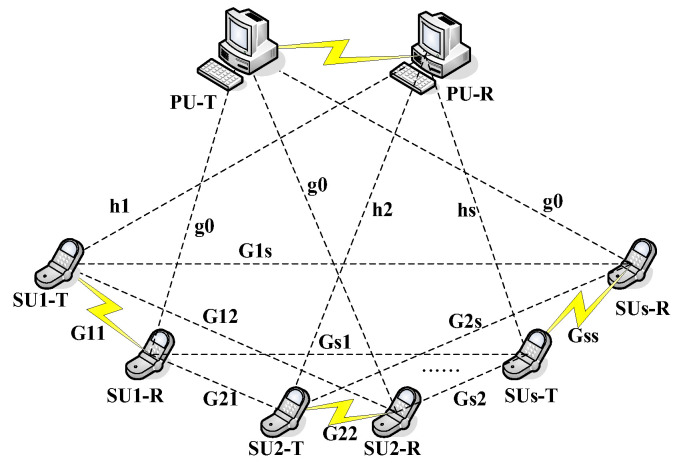
CR system model.

**Figure 2 sensors-22-06796-f002:**
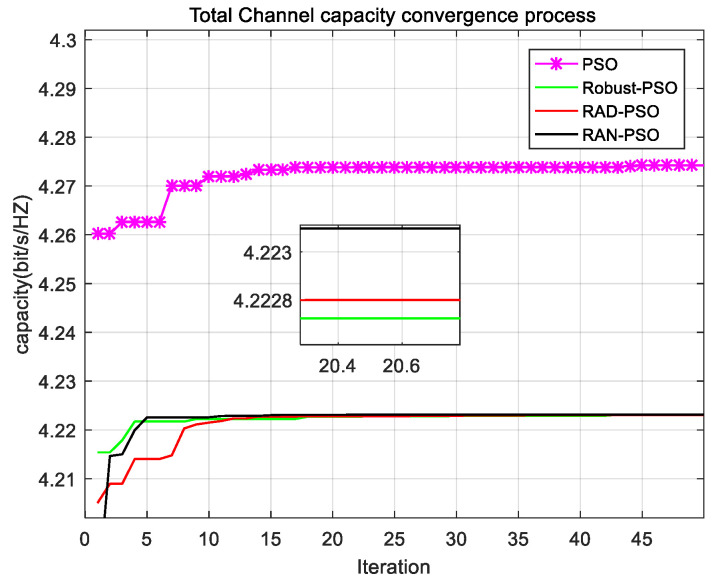
Convergence process of total channel capacity of cognitive users.

**Figure 3 sensors-22-06796-f003:**
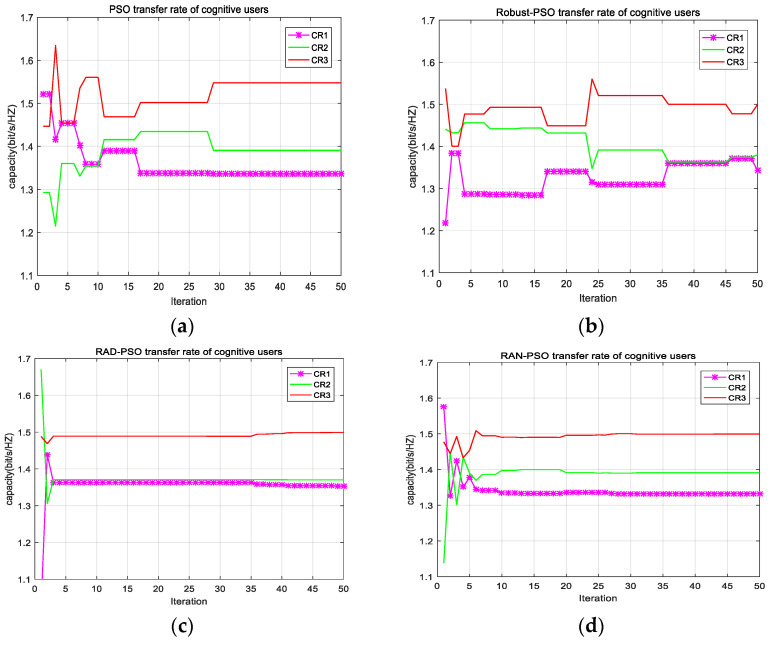
(**a**) Transmission rate of PSO algorithm; (**b**) transmission rate of Robust-PSO algorithm; (**c**) transmission rate of RAD-PSO algorithm; (**d**) transmission rate of RAN-PSO algorithm.

**Figure 4 sensors-22-06796-f004:**
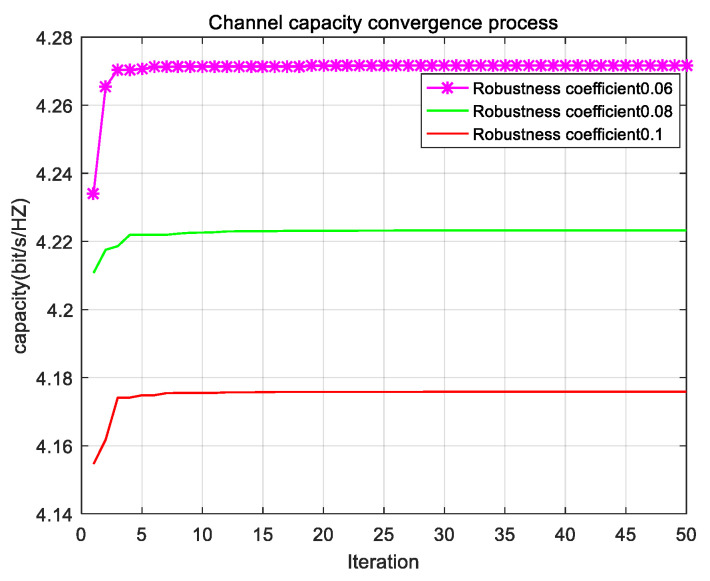
Convergence of channel capacity under different robust coefficients.

**Figure 5 sensors-22-06796-f005:**
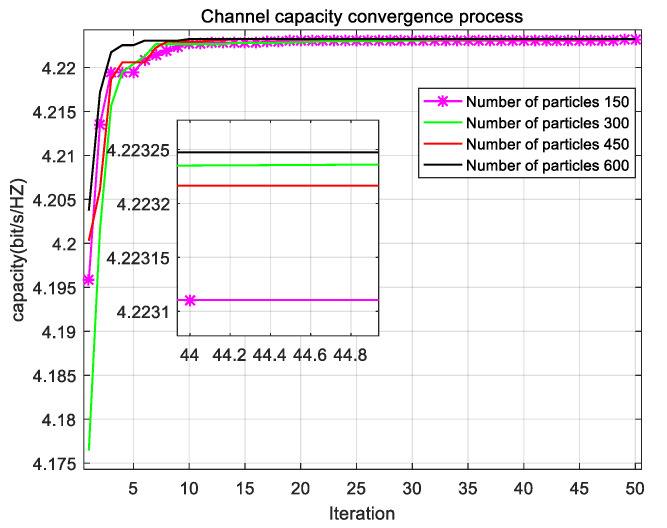
Comparison of channel capacity for different population sizes.

**Figure 6 sensors-22-06796-f006:**
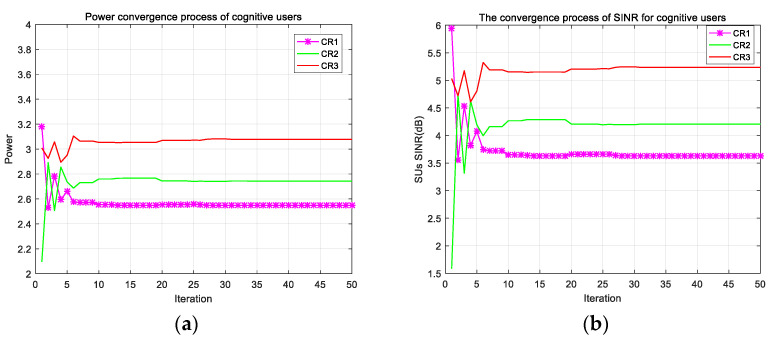
(**a**) Power convergence process of SUs; (**b**) SINR convergence process of cognitive users.

**Figure 7 sensors-22-06796-f007:**
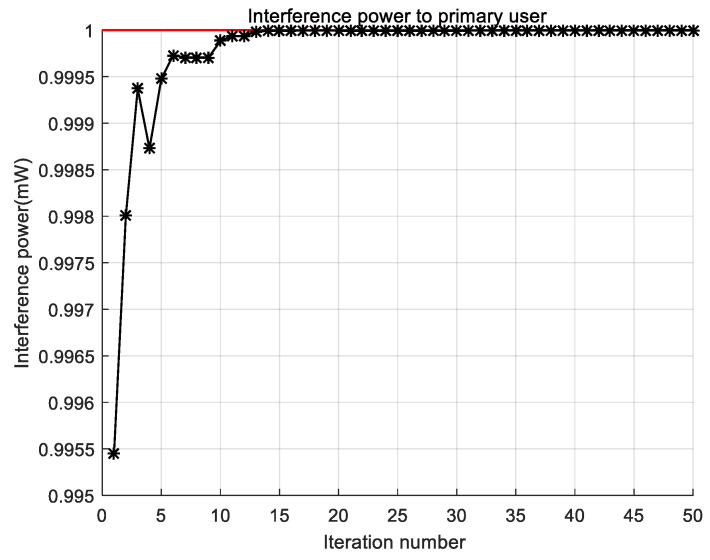
Cognitive user interference power to primary user.

**Table 1 sensors-22-06796-t001:** Robust model parameter setting.

Parameter	Size
Number of SU:	3
Number of PU:	1
Interference threshold of PU:	1 mw
Robustness coefficient:	0.08
Maximum transmission power of SU:	15 mw
transmission power of PU:	20 mw
Background noise:	0.1 mw
Transmission rate variance constraint of SU:	0.1

**Table 2 sensors-22-06796-t002:** Robust model parameter setting.

	PSO	Robust-PSO	RAD-PSO	RAN-PSO
Self-learning factor (*c*_1_):	0.8	0.8	0.8	0.8
Group learning factor (*c*_2_):	0.8	0.8	0.8	0.8
Inertia weight (*w*):	0.8	0.8	\	\
Population size:	300	300	300	300
Iterations:	50	50	50	50
Search space dimensions (D):	3	3	3	3
Inertia weight max (*w*_max_):	\	\	0.8	0.8
Inertia weight min (*w*_min_):	\	\	0.6	0.6

## Data Availability

Not applicable.

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
