# Peer review of "A Robust Power Allocation Algorithm for Cognitive Radio Networks Based on Hybrid PSO"

_sensors, 2022, doi:10.3390/s22186796_

Round 1
Reviewer 1 Report
Authors explained the "A Robust Power Allocation Algorithm for Cognitive Radio Net- 2 works Based on Hybrid PSO" where the study establishes a robust model which adds 13 the cognitive user transmission rate variance constraint to solve the maximum channel capacity time power allocation scheme by considering the worst-case channel transmission model, and finally solves this complex non-convex optimization problem by using the hybrid particle swarm algorithm.
Some points of corrections are still required like: How the proposed work is different from "Hybrid PSO-GSA for energy efficient spectrum sensing in cognitive radio network"
Every figure should be explained. Like Fig. 1 should be explained. Formatting of tables is very bad. Caption of figures is repetetive like in Fig. 4
I am not satisfied with the explaination given w.r.t. the results. It should be detailed much.
A comparison table of some related and latest work is also expected that may consider a) A Sequential Ensemble Model for Photovoltaic Power Forecasting b) A range based node localization scheme with hybrid optimization for underwater wireless sensor network c) Prosodic Feature-Based Discriminatively Trained Low Resource Speech Recognition System
English and Grammer needs to be corrected likie in "Research in the ad-hoc [10] CR network, S pairs of links for cognitive user transmission and a pair of links for primary user transmission are present, as shown in Fig.1". Avoid long sentences, as it creates hinderance in understanding the paper.
Reviewer 2 Report
This paper analyzes a robust optimization-based design for a cognitive wireless model. There are a number of issues in this work that need improvement and/or clarification:
1) Despite the emphasis on the cognitive feature of the model, the primary users are not taken into account at all, only through the interference threshold I which is totally external since the objective function depends only on the sum-rate of the SUs. Thus, the cognitive properties of the model are actually ignored.
2) The robust technique used in the optimization is to relax the constraints so that a worst-case model is solved.It is not clear how the uncertainty ellipsoids are calculated, and it is not clear to me that the original optimization problem has infinite constraints either. Please, clarify this point carefully.
3) Some of the parameters use in the simulation are not very realistic. Wireless devices use typically 10-20 dBm transmission power, for instance. While this does not change significantly the main conclusions, the simulations could have employed values better aligned with the current technology.
4) It is recommended that the text of the manuscript is thoroughly revised by a native English speaker.
Round 2
Reviewer 2 Report
The authors have addressed properly my previous concerns on this manuscript. The improved version fills the gaps identified concerning the model. As a consequence, the paper is recommended for publication.